# The Micromechanical Properties and Surface Roughness of Orthodontic Retainer Wires—An In Vitro Analysis

**DOI:** 10.3390/ma17143431

**Published:** 2024-07-11

**Authors:** Maciej Jedliński, Jolanta Krupa, Joanna Janiszewska-Olszowska

**Affiliations:** 1Department of Interdisciplinary Dentistry, Pomeranian Medical University in Szczecin, 70-111 Szczecin, Poland; joanna.janiszewska.olszowska@pum.edu.pl; 2Department of Machine Design and Maintenance, Faculty of Mechanical Engineering and Robotics, AGH University of Krakow, al. A. Mickiewicza 30, 30-059 Krakow, Poland

**Keywords:** retainers, orthodontics, hardness, roughness, elasticity, wire, stiffness, in vitro study

## Abstract

Background: Despite the large variety of retainer wires available, no studies could be found comparing the micromechanical properties and surface roughness of different retainer wires. Such characteristics affect the survival of the fixed retainer in terms of both fracture resistance and resistance to debonding from the tooth. Therefore, the aim of the present study was to examine and compare those characteristics in popular retainer wires. Methods: six different popular orthodontic retainer wires were subjected to instrumental indentation based on the Oliver and Pharr method. The geometric surface structure was analysed using a non-contact profilometer. Results: stainless steel wires had a higher hardness and a higher elastic modulus compared to titanium wires and white gold chain. The titanium wire and the white gold chain showed much more roughness than other wires. Conclusions: stainless steel wires are the most resistant, considering both the shape retention capacity and the ability to resist abrasive wear. The titanium wire showed the lowest hardness and, thus, the highest susceptibility to deformation. Bond-a-braid, Retainium and Orthoflex white gold are more resistant to fracture than other steel wires. Titanium wire and chain retainer wires have more roughness, which is a great advantage in terms of mechanical adhesion to composite materials.

## 1. Introduction

Orthodontic retention is an integral phase of orthodontic treatment, with the goal of maintaining the position of the teeth achieved during the active phase of treatment [1]. This aim is achieved through fixed and removable orthodontic retention [2]. The survival of fixed orthodontic retainers may be influenced by various factors, including natural mobility of the teeth bonded to the retainer under occlusal interferences [3], a tendency to relapse [4], occlusal settling during orthodontic retention [5] and forces related to food intake [6]. Nevertheless, it should be emphasized that the fixed retainer must be placed to omit all possible occlusal contacts in centric occlusion [7]. However, the influence of the occlusal forces on the fixed retainer is unavoidable. Whether the fixed retainer would be able to withstand various acting forces depends on the mechanical properties of the wire and the mechanical properties of the composite used for wire bonding. Recent scientific evidence has shown that there is no clinical superiority of one type of wire over another in terms of maintaining tooth position and improving the survival of fixed retainer [1,8]. However, this may be due to the fact that forces acting on a properly positioned fixed retainer are proportionally small and long term in nature. Surface interference during settling or the masticatory process requires the wire material to be hard enough to resist deformation, and elastic enough not to break in case of short concentration of force on a small part of the wire. What is more, roughness, geometry and surface structure of the wire is crucial for high retention of the composite which adheres to the wire solely by mechanical means. However, over time, each wire undergoes an increase in roughness due to conditions in the oral cavity [9], as well as its suspected tendency to age faster due to its shape and structure (braided vs. homogeneous) [10]. Any susceptibility to mechanical damage can lead to increased roughness and corrosion, increasing dental plaque and the accumulation and formation of calculus. Moreover, this may increase the probability of both gradual breakage—single strings of the retainer or the loosening of the wire structure—or complete breakage between the teeth.

Despite the high variety of retainer wires available, no studies could be found comparing the micromechanical properties and surface roughness of different retainer wires, which affects the survival of the fixed retainer in terms of both fracture resistance and resistance to debonding from the tooth. Nevertheless, the choice of an optimal wire seems crucial to the success of treatment.

The hardness and Young’s modulus are the primary characteristics that characterize the mechanical properties of materials. Hardness is a measure of the resistance to local plastic deformation induced by either mechanical indentation or abrasion. Young’s modulus is a measure of the stiffness of a solid material. The higher Young’s modulus is, the more stress is required to produce the same amount of strain. Roughness is a characteristic of surface texture. It can be measured by the amount by which the surface deviates from its ideal shape. If these deviations are large, the surface is rough; if they are small, the surface is smooth. Research on these characteristics has been carried out on orthodontic wires [11,12,13] or orthodontic brackets [14]. However, no studies were found that determined such characteristics for retainer wires.

Thus, the aim of the present study was to examine and compare the micromechanical properties and surface roughness of selected commercially available wires for fixed orthodontic retention.

## 2. Materials and Methods

### 2.1. Study Material

The study materials in the present investigation consisted of six different popular wires used for fixed orthodontic retainers according to the recent questionnaire study [15]:0.254 × 0.711 mm (0.010″ × 0.028″) stainless steel flat lingual retainer wire (RMO, Denver, CO, USA); (sample A);0.254 × 0.711 mm (0.010″ × 0.028″) braided stainless steel retainer wire (Ortho Technology, Tampa, FL, USA); (sample B);0.406 × 0.559 mm (0.016″ × 0.022″) braided stainless steel retainer wire “Bond-a-braid” (Reliance, Itasca, IL, USA); (sample C);0.965 × 0.381 mm (0.038″ × 0.015″) stainless steel chain “Ortho-flextech” wire (Reliance, Itasca, IL, USA); (sample D);0.279 × 0.686 mm (0.011″ × 0.027″) single strand lingual titanium wire “Retanium” (Reliance, Itasca, IL, USA); (sample E);0.965 × 0.381 mm (0.038″ × 0.015″) white gold chain “Ortho-flextech” wire (Reliance, Itasca, IL, USA); (sample F).

Four sections of wire were selected from each type and fixed in separate holders.

### 2.2. Determination of Micromechanical Properties

The hardness and modulus of elasticity tests of the retention wires were carried out by the instrumental indentation method based on the ISO 14577-1 [16] standard using the Nano-Hardness-Tester (S/N 1000110102, NHT^3^, Anton Paar, Gratz, Austria) with Step 500 platform.

The objects of study were retention wires, which are described in Section 2.1. The received wires were cut into segments of 15 mm in length and then permanently fixed in a unique holder adapted to fix them onto the NHT^3^ device. Indentation tests were performed in the center of the wider surfaces of the wires. Before measurements were performed, the surfaces were washed with isopropanol and allowed to dry.

The experiment consisted of loading a diamond indenter of specified geometry onto the surface of the sample with a linearly increasing force from 0 to a maximum fixed load *F_max_*, and simultaneously measuring the depth of penetration (h), which was recorded during the test for all of those values. Indentation tests were performed with a Berkovich diamond indenter using the following parameters:Maximum indenter load (*F_max_*): 25 mN;Loading/unloading speed: 50 mN/min;Load holding time at maximum load of 5 s.

At least 15 measurements were made in selected areas of each component, changing the measurement location by tens of micrometres each time to be free of the influence of deformation from the previous measurement. On the basis of the measurements, a load-displacement curve was plotted, which is used in the analysis of the elastic–plastic properties of the material tested according to the method of Oliver and Pharr [17,18]. Indentation hardness (*H_IT_*) was calculated according to the classical definition as the quotient of the indenter loading force and the imprint area according to the relation [17]:(1)$$H_{IT}=\frac{F_{max}}{A(h_{c})}$$
where *F_max_*—maximum load of the indenter, *A*(*h_c_*)—area of contact between the indenter and the sample at maximum load and *h_c_*—the depth along which contact is made between the indenter and the specimen.

Analysis of the strain of the indentation test according to Oliver and Pharr [18] makes it possible to calculate the elastic modulus of the material tested, often referred to as the indentation elastic modulus (*E_IT_*). The elastic modulus is calculated from the contact stiffness determined directly from the obtained indentation load-displacement curve according to the relation [17]:(2)$$E_{r}=\frac{\sqrt{\pi }\cdot S}{2\cdot \sqrt{A(h_{c})}}$$
where *S*—contact stiffness (tangent of the slope angle of the unloading curve); *A*(*h_c_*)—contact area considering permanent deformation and *E_r_*—reduced modulus of elasticity equal to:(3)$$\frac{1}{E_{r}}=\frac{1-{\vartheta_{s}}^{2}}{E_{IT}}+\frac{1-{\vartheta_{i}}^{2}}{E_{i}}$$
*E_IT_*—indentation elastic modulus, *ϑ_s_*—Poisson’s ratio of the tested material and *E_i_*, *ϑ_i_*—elastic modulus, Poisson’s ratio of the indenter material (for diamond *E* = 1141 GPa, *ν* = 0.07).

### 2.3. Geometric Structure of the Surface

Studies of the geometric surface structure analysis of the of the retention wires were performed using non-contact, optical profilometer (S/N 17G010, Profilm3D, Filmetrics, San Francisco, CA, USA). Measurements were taken in at least three locations on three different wires of each type on the flat lateral surface of the samples to determine the surface roughness described by Sa [µm] [19].

Arithmetical mean height (*S_a_*)—the average of the absolute elevation values (*Z*) at locations (*x*, *y*) within a defined assessment area, was calculated in accordance with ISO 25178-1 [20] as:(4)$$S_{a}=\frac{1}{A}\iint_{A}\left|Z\left(x,y\right)\right|dxdy$$
where *A* is the sampling area and *Z* is the absolute elevation values.

The measurements were made using 50× DI lens with a 4× digital zoom (Nikon, Tokyo, Japan). The measured working area was rectangular in shape with dimensions of 114 × 85 µm.

### 2.4. Statistical Analysis

Descriptive statistics for the measured characteristics (*E_IT_*, *H_IT_*, *h_max_*, *S_a_*) of six materials were calculated. Considering very different values of characteristic standard deviation in materials, the Kruskal-Wallis test [21] was used instead of ANOVA to access the significance of the difference between probes and the Wilcoxon rank sum post hoc test with Bonferroni corrections for multiple tests to determine which pairs of materials showed significant difference. The results were considered statistically significant at *p* < 0.05, where *p* value is the probability of obtaining test results equal to or more extreme than what was measured. The R statistical programme, ver. 4.3.0 (The R Foundation for Statistical Computing, Wirtschaftsuniversität Wien, Vienna, Austria) was used for statistical analyses.

## 3. Results

The indentation curves (Figure 1) illustrate the nature of the change in material deformation during indentation. Of particular interest is the section corresponding to unloading, where the contact stiffness (S) was determined according to Oliver and Pharr when the deformations were elastic in nature. Graphical representation of the load versus indentation allows for easy comparison of the deformation capacity of the materials tested. When comparing the following indentation curves, the stainless-steel chain wire (D) had the lowest indentation depth, which corresponded to the highest hardness value. The single strand titanium wire (E) showed the opposite properties—it had the lowest *H_IT_* and the highest *h_max_* value. Figure 1 shows the penetration pattern curve for each wire graph as a function of increasing the indenter load up to a maximum of 25 mN.

The results of the measurements of hardness (*H_IT_*), elastic modulus (*E_IT_*) and maximum penetration depth (*h_max_*) are summarized in Table 1, while Table 2 presents the roughness (*S_a_*) parameters. The scatter and distribution of the results are presented in Figure 2, Figure 3 and Figure 4 for micromechanical parameters, and in Figure 5 for roughness.

The Kruskal-Wallis test indicated statistically significant difference between materials in *H_IT_* (χ^2^ (5) = 51.044, *p* < 0.001), *E_IT_* (χ^2^ (5) = 47.09, *p* < 0.001), *h_max_* (χ^2^ [5] = 53.34, *p* < 0.001). Each chi-square distribution had 5 degrees of freedom. Wilcoxon rank sum post hoc test stated significant differences in some pairs of materials in each measured parameter.

The material F (gold chain) had significantly different distribution of *H_IT_*, with all five other wires—*p* < 0.001 while comparing to A, B, C, D (steel wires) and *p* = 0.008 while comparing to E (titanium wire) (Figure 2).

Wire B (braided stainless steel retainer wire, Ortho Technology) presented significantly larger values of *E_IT_* compared to D, E and F (stainless steel chain, Orthoflex; titanium wire; gold chain) as was *p* < 0.001 in all three cases. Material F (gold chain) presented significantly smaller values of *E_IT_* comparing to all samples but E (titanium wire) (*p* < 0.001 in all four cases), which can be seen in Figure 3.

For E wire (titanium wire), as with *H_IT_* and *E_IT_* values, *h_max_* had the largest scatter of results, in contrast to material F (gold chain) (*p* = 0.421).

The Kruskal-Wallis test indicated statistically significant difference in roughness between materials (χ^2^ (5) = 44.63, *p* < 0.001). The Wilcoxon rank sum post hoc test stated significant differences—wire E (titanium wire) had significantly different distribution of *h_max_* with all five other materials—*p* < 0.001 as compared to wires of the B, C and D (braided stainless steel retainer wire, Ortho Technology; braided stainless steel retainer wire “Bond-a-braid”, Reliance and stainless steel chain Orthoflex-tech, Reliance) and *p* = 0.006 as compared to A (stainless steel flat lingual retainer wire, RMO) (Figure 5).

Figure 6 presents the 3D structure surface of selected wires which were examined to measure *S_a_* roughness parameters.

The lowest roughness, at *S_a_* = 0.033 µm, was exhibited by material B—braided stainless steel retainer wire (Ortho Technology). A significantly higher Sa value was recorded compared to the other tested wires for sample E—single strand lingual titanium wire “Retanium” (Reliance), where *S_a_* = 1.126 µm. A heterogeneous geometric surface structure is characterized by the F wire—the white gold chain “Ortho-flextech” wire (Reliance), which had a deviation of 0.086 with an average roughness value of *S_a_* = 0.108 µm.

## 4. Discussion

Measurements made by the instrumental method made it possible to determine properties on the surface layer of materials, thin coatings and monolithic materials. Indentation measurements made at the load of 25 mN allowed for the determination of the hardness and elastic modulus on the surface layer of the tested wires. At a load of *F_max_* = 25 mN, the depth of penetration averaged 500 nm. In general, it could be concluded that steel wires (A, B, C and D) had higher hardness and elastic modulus compared to titanium and gold wires (E and F), which is consistent with the result of a recent study by Namura et al. [22] who also identified steel wires as the most resistant under in vitro conditions, and titanium wire as the least resistant but still clinically acceptable. The moduli of elasticity of steel wires were of a similar level and characteristic of steel (210 GPa). Classic metal wear models indicated that an increase in the hardness of the friction pair material leads to an increase in wear resistance. Wires with higher hardness experience less frictional wear. At the same time, a lower modulus of elasticity causes greater deformation in contact with hard particles when chewing food.

Retanium and Orthoflex White Gold (WG) wires have significantly lower hardness and a lower modulus of elasticity. The present finding that titanium wire has the lowest hardness, but at the same time the highest roughness of surface, may affect the measurement results. Thus, Retanium and Orthoflex WG are less prone to break, but are more prone to deform. This may depend on lower values of the elastic modulus and, consequently, lower stiffness, which makes the material more deformable and under the occlusal forces, it can deform plastically. If such a deformation were to occur under small, long-lasting forces, the position of the teeth may slightly change, although the retainer would still be bonded intactly to the teeth.

The highest penetration depth was found in titanium wire, gold chain and Bond-A-Braid steel wire. Clinically, such results may suggest that Bond-A-Braid, Retanium and Ortho-flextech WG are more resistant towards to breakage caused by occlusal loading than the other wires included in the study. However, there is no clinical evidence to support this statement [1,23]. Moreover, the shape of the wire is also an important factor when it comes to resistance to external forces. The highest hardness of 4.97 GPa was measured for Ortho-flex stainless steel chain wire. However, this finding was not concurrent with a higher modulus of elasticity, which may be partly due to shape. The load-carrying capacity is reduced in chain wire due to the smaller cross-section than in flat lingual wire or braided wire. Interestingly, Sifakakis et al. [24] pointed out in their study that Orthoflex White Gold is characterized by lower residual force after external loading than round stainless-steel wires, concluding that such a retainer is less prone to causing long-term inadvertent tooth movement than other tested wires. However, the residual loading decreased with the cross-section of the steel wire. In the co-authored book [25], this author concluded that “it is advisable to use wires with greater bending and torsional stiffness for fixed retention”, both to avoid too much deformation and to avoid the eventual storage of too much residual force. However, it should be noted that, as expressed by Arnold et al. [26], steel braided quadrangular wires have much better torque control than round wires. Thus, the present study did not analyze round retention wires.

It might be supposed that an interplay should exist between the mobility of the teeth, occlusal forces, the elasticity of the wire and the composite used to bond the wire to the teeth. In the contact interaction between elements, it is important to determine the geometric structure of the surface. Engler et al. [27] included microscopic images of fixed retainers at lower magnification but did not elaborate on the effect of the retainer surface on the bond strength to the tooth. It should be noted that the titanium wire exhibited significantly more roughness than the other wires analysed. Interestingly, it was followed by both wires in the Orthoflex series, first the steel wire, then the gold wire. The other three steel wires were characterized by a much lower roughness, of which the Bond-A-Braid had the highest Sa value. This seems important as the structure of the retainer wire should provide mechanical retention for the composite, which keeps it bonded to the teeth [28], a finding that has been evident in many studies on orthodontic brackets [29,30]. This may be an important reason for the long-term prognosis of fixed retainers, which is reflected in clinical studies. Kocher et al. indicate in their retrospective study that titanium wires debonded several per cent less frequently than braided steel wires [31]. Furthermore, Al-Maaitach et al. 2023 indicated in their study that Ortho-flex White Gold maintains tooth position better in the short term and detaches from the teeth 8% less frequently than multistranded steel wire [32]. Reichender et al. in their study on different types of fixed retainers noted that Bond-A-Braid had the highest shear bond strength of the steel and fibre-reinforced composite retainers they studied [33]. Their study did not include titanium retainer and Orthoflex wires, but it remained consistent with the results of the present study. In the end, it must be underlined that the macrostructure of chain and braided wires is also favorable for better composite adhesion, as both types of wire increase the number of ridges into which the composite can flow, thereby increasing the surface area of adhesion.

Interestingly, in a questionnaire study among Polish clinicians, it was the gold chain that was less likely to debond and deform, and the braided stainless-steel wire that was the easiest to bend, bond and the most effective at maintaining its shape [15]. Although the results of this in vitro study may indicate greater resistance to debonding, nevertheless, such wire may not be able to maintain its shape. This study also brought to light the evidence that steel, gold and titanium wires are the most popular among Polish orthodontists.

On the other hand, it has been discussed whether the higher roughness of orthodontic wires may be correlated with microbial adhesion. This can be especially important in fixed retainers as they stay in the mouth for a long period on the lingual surfaces of the anterior teeth, which makes daily hygiene in this area challenging [34]. Some studies provided evidence that the roughness of orthodontic wires is positively correlated with higher microbial adhesion [35,36], while others have strongly opposed this view [37]. However, recent in-depth research by Radovic et al. [38] proved that wire surface roughness was indeed correlated with microbial adhesion; however, it had the lowest influence on bacterial colonization of any orthodontic appliance.

The limitations of the present study arise from its nature. The wires studied may perform slightly differently in an oral cavity environment. In addition, other types of wire may be considered in future studies. The limitation of the method applied can be associated with the sensitivity to surface topography—a high value of roughness can affect the readings, which should be minimized by the proper selection of the material samples and the indenter. Furthermore, it may be interesting to validate the results of the present study through a tension load examination, an adhesion test between composite and wire, and a long-term future clinical trial.

## 5. Conclusions

(1)The most resistant wires are stainless steel braided Ortho technology wire and Orthoflex stainless steel chain wire, considering both shape retention capacity and the ability to resist abrasive wear. Ortho technology wire probably would retain the position of the teeth most effectively due to the highest stiffness and elastic modulus, as well as high hardness value.(2)Titanium wire showed the worst mechanical properties—it is the softest one, therefore the most susceptible to deformation. A low hardness value indicates that the wire might undergo both abrasion and permanent deformation more easily.(3)Bond-A-Braid, Retanium and Orthoflex WG are more resistant to fracture than other steel wires.(4)Titanium wire and chain wires have a much rougher structure than stainless steel wires, which is a great advantage in terms of proper mechanical adhesion to the composite in which the wire is embedded.

## Figures and Tables

**Figure 1 materials-17-03431-f001:**
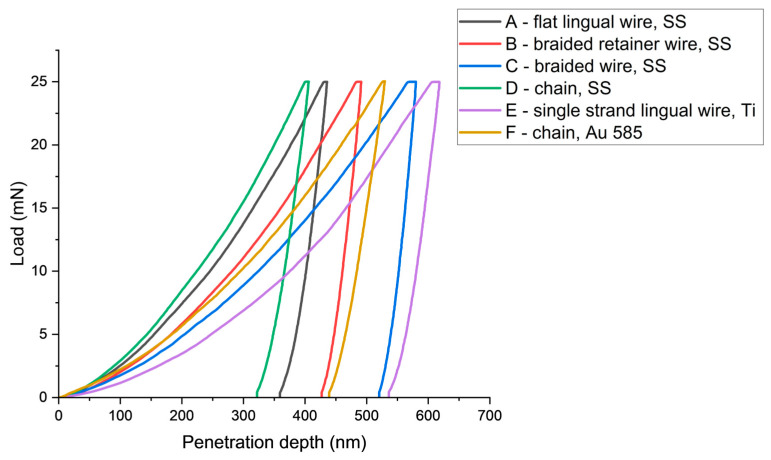
Plot of penetration depth (h) with increasing load (F) for each of the tested materials.

**Figure 2 materials-17-03431-f002:**
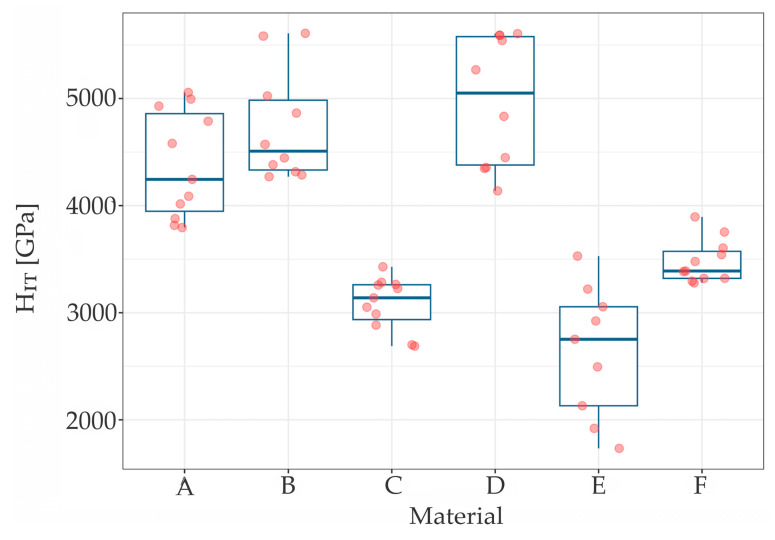
The H_IT_ distribution for tested samples.

**Figure 3 materials-17-03431-f003:**
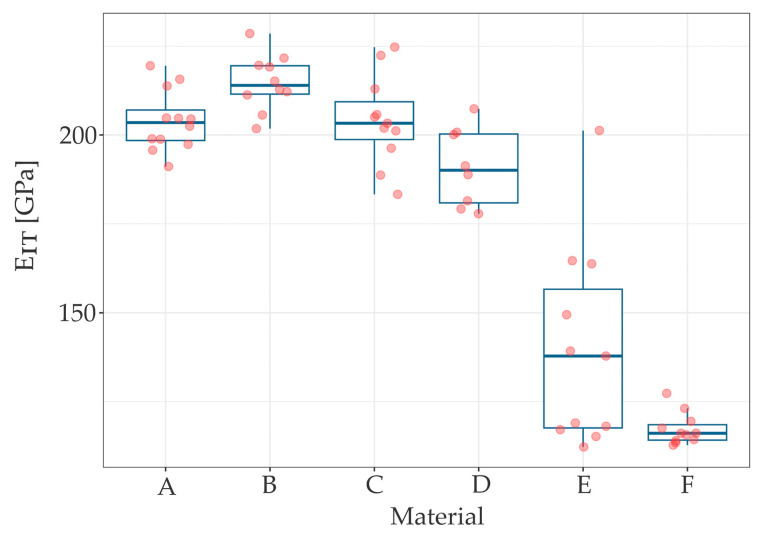
The E_IT_ distribution for tested samples.

**Figure 4 materials-17-03431-f004:**
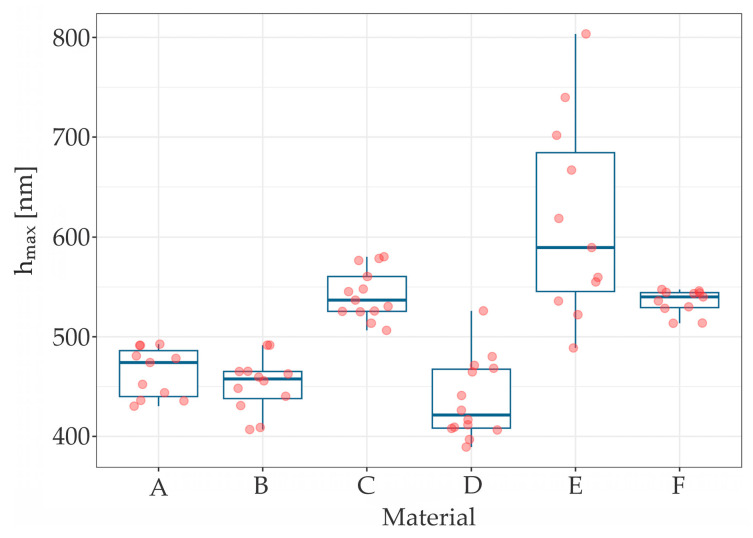
The *h_max_* distribution for tested samples.

**Figure 5 materials-17-03431-f005:**
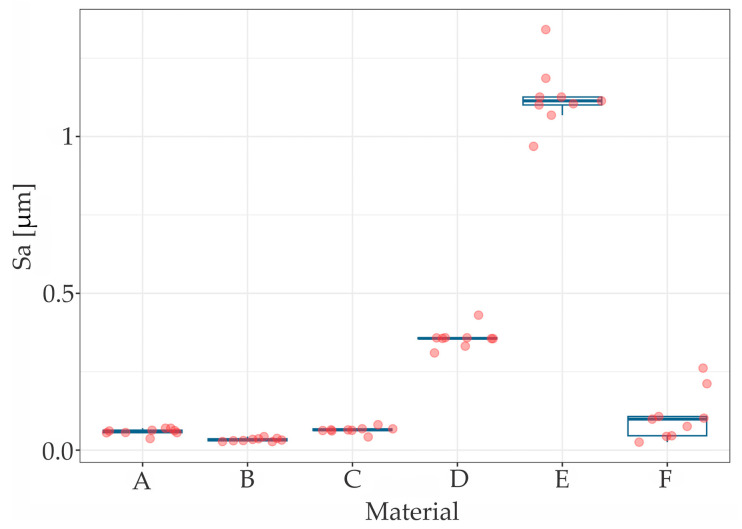
The *S_a_* distribution for tested samples.

**Figure 6 materials-17-03431-f006:**
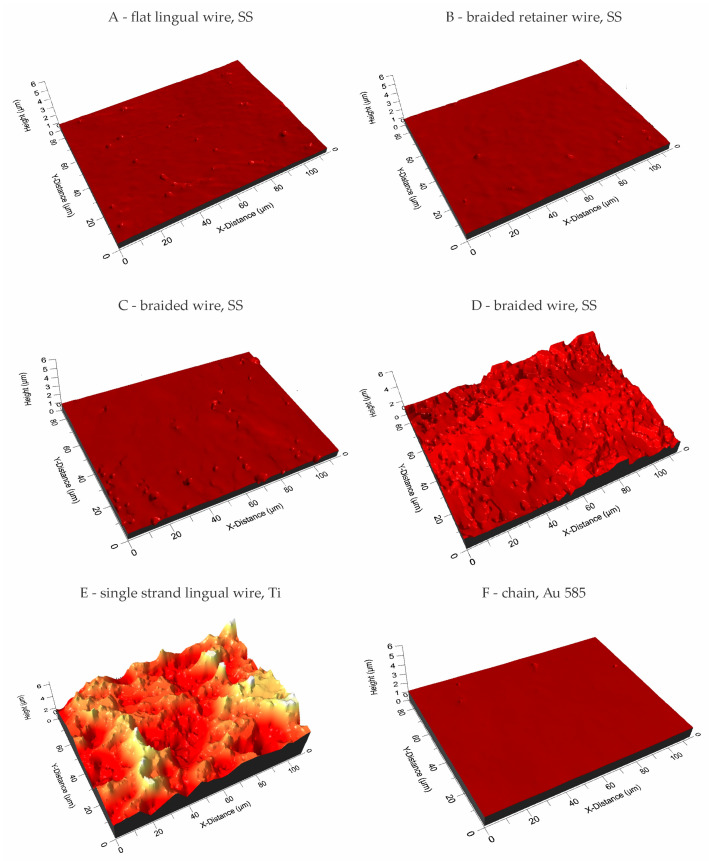
3D images of the surface of the tested wires.

**Table 1 materials-17-03431-t001:** Descriptive statistics of elastic modulus (*E_IT_*), hardness (*H_IT_*) and maximum penetration depth (*h_max_*). The size of the studied population *N* = 15; Min, Max and Mean are the minimum, maximum and mean values for a given sample; SD—standard deviation.

	Material	Min	Median	Max	Mean	SD
*H_IT_* [GPa]	A	3.80	4.25	5.06	4.38	0.48
	B	4.27	4.51	5.61	4.74	0.49
	C	2.69	3.14	3.43	3.08	0.23
	D	3.80	4.25	5.06	4.38	0.56
	E	1.73	2.75	3.53	2.64	0.58
	F	3.28	3.39	3.89	3.48	0.19
*E_IT_* [GPa]	A	191.11	203.49	219.51	203.95	8.2
	B	201.79	213.98	228.59	214.79	7.5
	C	183.26	203.31	224.76	204.14	12.0
	D	177.82	190.07	207.38	190.85	11.2
	E	112.21	137.83	201.26	139.80	26,7
	F	112.73	116.07	127.30	117.23	4.3
*h_max_* [nm]	A	430.35	474.12	492.60	464.25	24.83
	B	406.92	457.64	491.59	452.26	27.22
	C	506.40	536.77	580.32	542.53	24.89
	D	389.34	421.46	526.00	436.86	39.53
	E	488.76	589.51	803.54	616.52	99.73
	F	513.57	540.00	547.46	535.20	12.33

**Table 2 materials-17-03431-t002:** Descriptive statistics of surface roughness values (*S_a_*). The size of the studied population *N* = 9.

	Material	Min	Median	Max	Mean	SD
*S_a_* [µm]	A	0.037	0.061	0.070	0.059	0.016
	B	0.027	0.032	0.043	0.033	0.007
	C	0.042	0.065	0.081	0.064	0.016
	D	0.310	0.356	0.430	0.357	0.052
	E	0.969	1.114	1.341	1.126	0.157
	F	0.025	0.099	0.262	0.108	0.086

## Data Availability

The raw data are available upon reasonable request by corresponding with the author.

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
