# Peer review of "The Micromechanical Properties and Surface Roughness of Orthodontic Retainer Wires—An In Vitro Analysis"

_materials, 2024, doi:10.3390/ma17143431_

Round 1
Reviewer 1 Report
Comments and Suggestions for Authors
The manuscript describes the investigation of the micro-mechanical properties and surface roughness of 6 different commercial retainer wires.
The topic and scope suit the special issue “Mechanical Behavior of Composite Materials”, and the aspects encompassing micro-mechanical properties and surface roughness are comprehensive.
Overall, the manuscript needs to be revised to improve reproducibility and result dissemination. Details are as following:
The reproducibility needs to be improved:
- the wire dimensions should also be stated in ISO unit of mm
- full product information should be given for each instrument in the format (serial No., brand, manufacturer, city, country)
- statistical analysis section is missing. variance test should be done to test if the difference of mean amongst groups is statistically significant or not and the results should also be added to Table 1 and Fig. 2. Report p values where appropriate.
Manuscript needs to be edited, pay attention to the format of all equations, especially the subscripted part, also any symbol with subscripted part in the text. Organization is not perfect, for example Line118-122 this section should be moved up to before the indentation test.
Table 1, H values please also use GPa.
Table 2, maybe a fig. is better and please increase the readability
Line 66 after Research on these characteristics has been carried out on orthodontic wires [11,12], Suggest citing also the publication that studied all three characteristics of archwires: Biomedical NiTi and β-Ti Alloys: From Composition, Microstructure and Thermo-Mechanics to Application.
Instrumental indentation based on Oliver and Pharr method is not commonly used for the determination of Young’s modulus of elasticity for orthodontic wires. Why did you use this method? Of course it has the advantage of ease of operation and surface hardness can be obtained simultaneously. Please discuss the limitation of this convenient method, the assumptions it makes and if these apply to the wire sample here. Also compare the results obtained with conventional methods such as tensile test, 3-point bending test and cantilever bending, to shed light on if this method is applicable to orthodontic wires in general.
Extensive text editing should be carried out to correct English usage such as “The wires in the state of delivery” should be “The as received wires”; right after each equation, introduce parameters stating “Where A is…; B is…etc.; and Line 118 “in a section 3.1” which does not exist. There are more.
Comments on the Quality of English Languageminor check
Author Response
Dear Reviewer,
thank You for the positive reception of our manuscript. Please find the answers to Your queries in the italics and changes within the manuscript in red.
- The reproducibility needs to be improved:
the wire dimensions should also be stated in ISO unit of mm
full product information should be given for each instrument in the format (serial No., brand, manufacturer, city, country)
-Table 1, H values please also use GPa.
Table 2, maybe a fig. is better and please increase the readability
Manuscript needs to be edited, pay attention to the format of all equations, especially the subscripted part, also any symbol with subscripted part in the text. Organization is not perfect, for example Line118-122 this section should be moved up to before the indentation test.
Thank you for these valuable comments. We have applied all the corrections You kindly suggested in the manuscript.
- statistical analysis section is missing. variance test should be done to test if the difference of mean amongst groups is statistically significant or not and the results should also be added to Table 1 and Fig. 2. Report p values where appropriate.
Thank you very much for your valuable comment on the issues of statistical analysis. In order to meet Your requirements firstly the authors has applied descriptive statistics, and then Kruskal-Wallis tests and Wilcoxon rank sum post-hoc test with Bonferroni corrections for multiple testing.
- Line 66 after Research on these characteristics has been carried out on orthodontic wires [11,12], Suggest citing also the publication that studied all three characteristics of archwires: Biomedical NiTi and β-Ti Alloys: From Composition, Microstructure and Thermo-Mechanics to Application.
Thank You for Your valuable suggestion. We added proper citation to the article.
- Instrumental indentation based on Oliver and Pharr method is not commonly used for the determination of Young’s modulus of elasticity for orthodontic wires. Why did you use this method? Of course it has the advantage of ease of operation and surface hardness can be obtained simultaneously. Please discuss the limitation of this convenient method, the assumptions it makes and if these apply to the wire sample here. Also compare the results obtained with conventional methods such as tensile test, 3-point bending test and cantilever bending, to shed light on if this method is applicable to orthodontic wires in general.
The authors agree that instrumental indentation based on Oliver and Pharr method is not commonly used when it comes to retainer wires, nevertheless, there can be found a few publications on the use of this method to mechanical properties investigation of the orthodontic wires, e.g. O. Annousaki, S. Zinelis, G. Eliades, T. Eliades, Comparative analysis of the mechanical properties of fiber and stainless steel multistranded wires used for lingual fixed retention, Dental Materials, Volume 33, Issue 5,2017, Pages e205-e211.
Moreover, our investigation of stainless steel wires was conducted in accordance with the ISO 14577 standard, where the measurement methodology is based on the instrumental indentation test for determination of the hardness and other material parameters of the metals. In this work, we focused particularly on the surface layer of these wires, hence we used a measurement method that allowed us to examine the mechanical properties in the surface layer of the wires.
Complementary testing of the surface layer, involving the determination of the mechanical properties and surface topography made it possible to identify those features of the materials that allow for ensuring the potentially highest adhesion of the wire-composite-tooth joint and the relatively highest abrasive resistance and plastic deformation. Hardness and modulus of elasticity have a significant impact on the contact interaction during the operation of wires permanently bonded to teeth.
Our overarching goal is to build a database of the presented materials for simulation behaviour of wire in contact interactions. Thank you for underlining the importance of the conventional mechanical testing. We plan to extend existing research with methodology you have mentioned: tensile and bending tests. The difference between the conventional tests and the Olivier and Parr method is that they describe the properties of the material as a whole, not just the surface layer, the characterization of which was the purpose of our work.
The limitation of this method can be the sensitivity to surface topography - a high value of roughness will affect the readings. Attention should be paid to the choice of indenter geometry - it depends on the type of material being tested. Based on the Oliver and Pharr method, hardness and Young's modulus are determined from indentation depth data for a load/unload cycle. When it comes to instrumented testing, micro and nano indentation are primarily designed for testing the surface layers of samples.
- Extensive text editing should be carried out to correct English usage such as “The wires in the state of delivery” should be “The as received wires”; right after each equation, introduce parameters stating “Where A is…; B is…etc.; and Line 118 “in a section 3.1” which does not exist. There are more.
Thank you very much for your detailed analysis of the text. We have corrected the errors pointed out and checked the text similarly correcting found errors.
Reviewer 2 Report
Comments and Suggestions for Authors
Dear Authors,
I have reviewed your manuscript titled " The micromechanical properties and surface roughness of orthodontic retainer wires – an in-vitro analysis." Overall, the article is well-written and and the research is conducted in accordance with scientific requirements. The study is extensive and well documented, the information is properly processed. I appreciate the fact that you specified the limits of this study.I commend you on your thorough research and clear presentation. However, I have a few suggestions for minor revisions that I believe will enhance the clarity and completeness of your manuscript.
1. Specifying manufacturers and trade names is not recommended
2. In my opinion, table 2 is actually figure 2
3. Are you thinking of continuing the study in vivo?
4.I suggest you review the References chapter once more in order to meet the requirements. Some articles have the year - Bold, others do not. Similarly, in some we find the title in Italics, in others not, e.a.
Thank you for your attention to these suggestions.
Best regards!
Author Response
Dear Reviewer,
thank You for the positive reception of our manuscript. Please find the answers to Your suggestions in italics and changes within the manuscript In red.
- Specifying manufacturers and trade names is not recommended
The authors agree that when it comes to characteristics of the tested materials the trade name is not recommended. However, we would like to address the article to wider audience, namely to both representatives of materials and mechanical engineering, as well as orthodontists who do not have detailed material knowledge as our recent questionnaire study proved. It has been cited under the number 15.
- In my opinion, table 2 is actually figure 2
Thank You for Your kind remark, we have corrected this issue – we added separated table with mean values and standard deviations for Sa parameters, so figure 2 was deleted.
- Are you thinking of continuing the study in vivo?
Thank You for the interesting suggestion. The aim of this article is to characterize the surface layer properties of the retention wires which in long term study we plan, is the first step. In the future the authors plan to examine wires under tension load as well as conduct adhesion test between composite and wire. To address Your comment, we added a proper sentence on the discussion: “Furthermore, it may be interesting to validate the results of the present study through tension load examination, an adhesion test between composite and wire as well as a long-term future clinical trial.”
4.I suggest you review the References chapter once more in order to meet the requirements. Some articles have the year - Bold, others do not. Similarly, in some we find the title in Italics, in others not, e.a.
Thank you very much for your detailed inspection. The Reference chapter has been corrected.
Reviewer 3 Report
Comments and Suggestions for Authors
Dear author, I congratulate you on a correctly developed study. I recommend to improve the text:
1. In the material and method, I would specify the reference to popularly used wires. Are they the ones that sell the most by price or the ones that have a greater indication?
2. In the results what refers to table 2: 3D images. I would leave it as graphics. And in the editing I would shrink them a little so that the 6 images would remain on one page.
3. In the discussion I would make a reference to the properties that exist in the clinical context in relation to their indication for the movement of teeth, for example, and not only in their resistance or probability of detachment.
Kind regards
Author Response
Dear author, I congratulate you on a correctly developed study. I recommend to improve the text:
Dear Reviewer, thank You for the positive reception of our manuscript. Please find reply to Your comments in italics and changes within the manuscript in red.
- In the material and method, I would specify the reference to popularly used wires. Are they the ones that sell the most by price or the ones that have a greater indication?
Thank You for Your kind comment. The wires were selected on the basis of a questionnaire study on a large group of Polish orthodontists, published in the Materials. The authors have now added a proper citation in the methods section, as: The study materials in the present investigation consisted of six different popular wires used for fixed orthodontic retainers in accordance with the recent questionnaire surveys [15]:
- In the results what refers to table 2: 3D images. I would leave it as graphics. And in the editing I would shrink them a little so that the 6 images would remain on one page.
Thank You for Your valuable suggestion. The authors has now changed the formatting of the manuscript to fit it on one page, the Table 2 became Figure 5.
- In the discussion I would make a reference to the properties that exist in the clinical context in relation to their indication for the movement of teeth, for example, and not only in their resistance or probability of detachment.
Dear reviewer, in the discussion the authors underline that the susceptibility to plastic deformation may be a source of tooth displacement, even if the retainer will be still bonded to the teeth. However, to convey this information in a direct way, we added a relevant sentence in the discussion.
Kind regards
Round 2
Reviewer 1 Report
Comments and Suggestions for Authors
All the issues have been addressed.